# GradSign: Model Performance Inference with Theoretical Insights

**Zhihao Zhang**
Carnegie Mellon University
`zhihaoz3@cs.cmu.edu`

**Zhihao Jia**
Carnegie Mellon University
`zhihao@cmu.edu`

## Abstract

A key challenge in neural architecture search (NAS) is quickly inferring the predictive performance of a broad spectrum of neural networks to discover statistically accurate and computationally efficient ones. We refer to this task as model performance inference (MPI). The current practice for efficient MPI is gradient-based methods that leverage the gradients of a network at initialization to infer its performance. However, existing gradient-based methods rely only on heuristic metrics and lack the necessary theoretical foundations to consolidate their designs. We propose GradSign, an accurate, simple, and flexible metric for model performance inference with theoretical insights. A key idea behind GradSign is a quantity $\Psi$ to analyze the *sample-wise optimization landscape* of different networks. Theoretically, we show that $\Psi$ is an upper bound for both the training and true population losses of a neural network under reasonable assumptions. However, it is computationally prohibitive to directly calculate $\Psi$ for modern neural networks. To address this challenge, we design GradSign, an accurate and simple approximation of $\Psi$ using the gradients of a network evaluated at a random initialization state. Evaluation on seven NAS benchmarks across three training datasets shows that GradSign generalizes well to real-world neural networks and consistently outperforms state-of-the-art gradient-based methods for MPI evaluated by Spearman's $\rho$ and Kendall's Tau. Additionally, we have integrated GradSign into four existing NAS algorithms and show that the GradSign-assisted NAS algorithms outperform their vanilla counterparts by improving the accuracies of best-discovered networks by up to 0.3%, 1.1%, and 1.0% on three real-world tasks. Code is available at `https://github.com/JackFram/GradSign`

## 1 Introduction

As deep learning methods evolve, neural architectures have gotten progressively larger and more sophisticated (He et al., 2015; Ioffe & Szegedy, 2015; Krizhevsky et al., 2017; Devlin et al., 2019; Rumelhart et al., 1986; Srivastava et al., 2014; Kingma & Ba, 2017), making it increasingly challenging to *manually* design model architectures that can achieve state-of-the-art predictive performance. To alleviate this challenge, recent work has proposed several approaches to *automatically* discovering statistically accurate and computationally efficient neural architectures. The most common approach is *neural architecture search* (NAS), which explores a comprehensive search space of potential network architectures that use a set of predefined network modules as basic building blocks. Recent work shows that NAS is able to discover architectures that outperform human-designed counterparts (Liu et al., 2018a; Zoph & Le, 2016; Pham et al., 2018).

A key challenge in NAS is quickly assessing the predictive performance of a diverse set of candidate architectures to discover performant ones. We refer to this task as *model performance inference* (MPI). A straightforward approach to MPI is directly training each candidate architecture on a dataset until convergence and recording the achieved training loss and validation accuracy (Frankle & Carbin, 2018; Chen et al., 2020a; Liu et al., 2018a; Zoph & Le, 2016). Though accurate, this approach is computationally prohibitive and cannot scale to large networks or datasets.

The current practice to efficient MPI is *gradient-based methods* that leverage the gradient information of a network at initialization to infer its predictive performance (Lee et al., 2018; Wang et al., 2020; Tanaka et al., 2020). Compared to directly measuring the accuracy of candidate networks on a training

dataset, gradient-based methods are computationally more efficient since they only require evaluating a mini-batch of gradients at initialization. However, existing gradient-based methods rely only on heuristic metrics and lack the necessary theoretical insights to consolidate their designs.

In this paper, we propose GradSign, a simple yet accurate metric for MPI with theoretical foundations. GradSign is inspired by analyzing the *sample-wise optimization landscape* of a network. GradSign takes as inputs a mini-batch of sample-wise gradients evaluated at a random initialization point and outputs a statistical evidence of a network that highly correlates to its well-trained predictive performance measured by accuracy on the entire dataset.

Prior theoretical results (Allen-Zhu et al., 2019) show that the optimization landscape of a randomly initialized network is nearly convex and semi-smooth for a sufficiently large neighborhood. To realize its potential for MPI, we generalize these results to sample-wise optimization landscapes and propose a quantity $\Psi$ to measure the density of sample-wise local optima in the convex areas around a random initialization point. Additionally, we prove that both the training loss and generalization error of a network are proportionally upper bounded by $\Psi^2$ under reasonable assumptions.

Based on our theoretical results, we design GradSign, an accurate and simple approximation of $\Psi$. Empirically, we show that GradSign can also generalize to realistic setups that may violate our assumptions. In addition, GradSign is efficient to compute and easy to implement as it uses only the sample-wise gradient information of a network at a random initialization point.

Extensive evaluation of GradSign on seven NAS benchmarks (i.e., NAS-Bench-101, NAS-Bench-201, and five design spaces of NDS) across three datasets (i.e., CIFAR-10, CIFAR-100, and ImageNet16-120) shows that GradSign consistently outperforms existing gradient-based methods in all circumstances. Furthermore, we have integrated GradSign into existing NAS algorithms and show that the GradSign-assisted variants of these NAS algorithms lead to more accurate architectures.

**Contributions.** This paper makes the following contributions:

- We provide a new perspective to view the overall optimization landscape of a network as a combination of sample-wise optimization landscapes. Based on this insight, we introduce a new quantity $\Psi$ that provides an upper bound on both the training loss and generalization error of a network under reasonable assumptions.

- To infer $\Psi$, we propose GradSign, an accurate and simple estimation of $\Psi$. GradSign enables fast and efficient MPI using only the sample-wise gradients of a network at initialization.

- We empirically show that GradSign generalizes to modern network architectures and consistently outperforms existing gradient-based MPI methods. Additionally, GradSign can be directly integrated into a variety of NAS algorithms to discover more accurate architectures.

## 2 RELATED WORK

### 2.1 MODEL PERFORMANCE INFERENCE

Table 1 summarizes existing approaches to inferring the statistical performance of neural architectures.

**Sample-based methods** assess the performance of a neural architecture by training it on a dataset. Though accurate, sample-based methods require a surrogate training procedure to evaluate each architecture. EconNAS (Zhou et al., 2020) mitigates the cost of training candidate architectures by reducing the number of training epochs, input dataset sizes, resolution of input images, and model sizes.

**Theory-based methods** leverage recent advances in deep learning theory, such as Neural Tangent Kernel (Jacot et al., 2018) and Linear Region Analysis (Serra et al., 2018), to assess the predictive performance of a network (Chen et al., 2020a; Mellor et al., 2021; Park et al., 2020). In particular, NNGP (Park et al., 2020) infers a network's performance by fitting its kernel regression parameters on a training dataset and evaluating its accuracy on a validation set, which alleviates the burden of training. As another example, Chen et al. (2020a) utilizes the kernel condition number proposed in Xiao et al. (2020), which can be theoretically proved to correlate to training convergence rate and generalization performance. However, this theoretical evidence is only guaranteed for extremely wide networks with a specialized initialization mode. While the linear region analysis used in Mellor

Table 1: A summary of existing methods for model performance inference. The right four columns show (1) whether a method is based on theoretical results, (2) whether a method avoids expensive training process, (3) whether a method is applicable to different model architectures, and (4) whether a method is applicable across different tasks.

| | Methods | Theoretical Insight | Training Free | Model Independent | Task Independent |
|---|---|---|---|---|---|
| **Sample-Based** | EconNAS | ✗ | ✗ | ✓ | ✓ |
| **Theory-Based** | NNGP, TE-NAS, NASWOT, ZenNAS | ✓ | ✓ | ✗ | ✓ |
| **Learning-Based** | Neural Predictor, One-Shot-NAS-GCN | ✗ | ✗ | ✓ | ✗ |
| **Gradient-Based** | Snip, Grasp, Synflow Fisher | ✗ | ✓ | ✓ | ✓ |
| | GradSign (this paper) | ✓ | ✓ | ✓ | ✓ |

et al. (2021), Lin et al. (2021) and Chen et al. (2020a) is easy to implement, such technique is only applicable to networks with ReLU activations (Agarap, 2018).

**Learning-based methods** train a separate network (e.g., graph neural networks) to predict a network's accuracy (Liu et al., 2018a; Luo et al., 2020; Dai et al., 2019; Wen et al., 2020; Chen et al., 2020b; Siems et al., 2020). Though these learned models can achieve high accuracies on a specific task, this approach requires constructing a training dataset with sampled architectures for each downstream task. As a result, existing learning-based methods are generally task-specific and computationally prohibitive.

**Gradient-based methods** infer the statistical performance of a network by leveraging its gradient information at initialization, which can be easily obtained using an automated differentiation tool of today's ML frameworks, such as PyTorch (Paszke et al., 2017) and TensorFlow (Abadi et al., 2016). The weight-wise salience score computed by several pruning at initialization (Lee et al., 2018; Wang et al., 2020; Tanaka et al., 2020) methods can easily be adapted to MPI settings by summing scores up. Though lack of theoretical foundations, such migrations have been empirically proven to be effective as baselines in recent works (Abdelfattah et al., 2021a; Mellor et al., 2021; Lin et al., 2021). An alternative stream of work (Turner et al., 2019; 2021; Theis et al., 2018) uses approximated second-order gradients, known as empirical Fisher Information Matrix (FIM), at a random initialization point to infer the performance of a network. Empirical FIM (Martens, 2014) is a valid approximation of a model's predictive performance only if the model's parameters are a Maximum Likelihood Estimation (MLE). However, this assumption is invalid at a random initialization point, making FIM-based algorithms inapplicable. A key difference between GradSign and existing gradient-based methods is that GradSign is based on a fine-grained analysis of sample-wise optimization landscapes rather than heuristic insights. In addition, GradSign also provides the first attempt for MPI by leveraging the optimization landscape properties contained in sample-wise gradient information, while prior gradient-based methods only focus on gradients evaluated in a full batch fashion.

## 2.2 NEURAL ARCHITECTURE SEARCH

Recent work (He et al., 2021; Cai et al., 2019; 2018; Tan & Le, 2019; Howard et al., 2019) has proposed several algorithms to explore a NAS search space and discover highly accurate networks. RS (Bergstra & Bengio, 2012) is one of the baseline algorithms that generates and evaluates architectures randomly in the search space. REINFORCE (Williams, 1992) moves a step forward by reframing NAS as a reinforcement learning task where accuracy is the reward and architecture generation is the policy action. Given limited computational resources, BOHB (Falkner et al., 2018) uses Bayesian Optimization (BO) to propose candidates while uses HyperBand(HB) (Li et al., 2017) for searching resource allocation. REA (Real et al., 2019) uses a simple yet effective evolutionary searching strategy that achieves state-of-the-art performance. GradSign is complementary to and can be combined with existing NAS algorithms. We integrate GradSign into the NAS algorithms mentioned above and show that GradSign can consistently assist these NAS algorithms to discover more accurate architectures on various real-world tasks.

## 2.3 OPTIMIZATION LANDSCAPE ANALYSIS

Inspired by the fact that over-parameterized networks always find a remarkable fit for a training dataset (Zhang et al., 2016), optimization landscape analysis has been one of the main focuses in

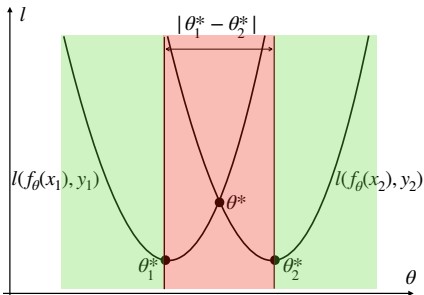
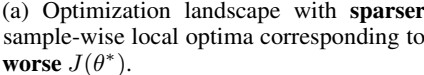
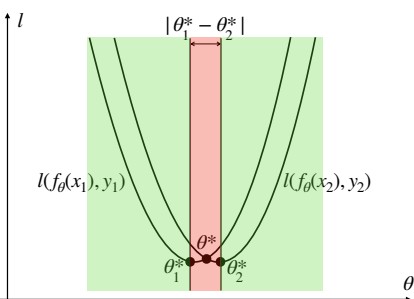

(a) Optimization landscape with **sparser** sample-wise local optima corresponding to **worse** $J(\theta^*)$.

(b) Optimization landscape with **denser** sample-wise local optima corresponding to **better** $J(\theta^*)$.

Figure 1: Illustration of our theoretical insight that denser sample-wise local optima indicate lower training losses. As the distances ($|\theta_1^* - \theta_2^*|$, shown in red) between the local optima across samples reduce, there is a higher probability that the gradients of different samples have the same sign at a random initialization point, shown as the green areas.

deep learning theory (Brutzkus & Globerson, 2017; Du et al., 2018; Ge et al., 2017; Li & Yuan, 2017; Soltanolkotabi, 2017; Allen-Zhu et al., 2019). Even though existing theoretical results for optimization landscape analysis rely on strict assumptions on the landscape's smoothness, convexity, and initialization point, we can leverage theoretical insights to guide the design of GradSign. In addition, SGD-based optimizers trained from randomly initialized points hardly encounter non-smoothness or non-convexity in practice for a variety of architectures (Goodfellow et al., 2014). Furthermore, Allen-Zhu et al. (2019) provides theoretical evidence that for a sufficiently large neighborhood of a randomly initialized point, the optimization landscape is nearly convex and semi-smooth. Different from existing optimization landscape analyses depending on objectives evaluated across a mini-batch of training samples, we propose a new perspective that decomposes a mini-batch objective into the aggregation of sample-wise optimization landscapes. To the best of our knowledge, our work is the first attempt to MPI by leveraging sample-wise optimization landscapes.

## 3 THEORETICAL FOUNDATIONS

### 3.1 INSIGHTS

Conventional optimization landscape analyses focus on objectives across a mini-batch of training samples and miss potential evidence hidden in the optimization landscapes of individual samples. By decomposing a mini-batch objective into the summation of *sample-wise* objectives across individual samples in a mini-batch, we can distinguish better local optima as illustrated in Figure 1. Both Figure 1a and Figure 1b reach a local optimum at $\theta^*$ for the mini-batch objective $J = \frac{1}{2}(l(f_{\theta^*}(x_1), y_1) + l(f_{\theta^*}(x_2), y_2))$. However, the optimization landscape in Figure 1b contains a better local optimum $\theta^*$ (i.e., a lower $J$). This can be distinguished by analyzing the relative distance between local optima across training samples (i.e., $|\theta_1^* - \theta_2^*|$ in Figure 1).

For a mini-batch with more than two samples, we use a sample-wise local optima density measurement $\Psi$ defined in Section 3.2 to represent the overall closeness of sample-wise local optima. Intuitively, as the distances between the local optima across samples reduce (shown as the red areas in Figure 1), there is a higher probability that the gradients of different samples evaluated at a random initialization point have the same sign (shown as the green areas in Figure 1). Driven by this insight, we propose GradSign to infer the sample-wise local optima density $\Psi$ statistically. The design of GradSign is based on our theoretical results that a network with denser sample-wise local optima has lower training and generalization losses under reasonable assumptions. We introduce the notations and assumptions in Section 3.2, provide a formal derivation of our theoretical results in Section 3.3, and present GradSign in Section 4.

### 3.2 PRELIMINARIES

We use $\mathcal{S} = \{(x_i, y_i)\}_{i \in [n]}$ to denote training samples, where each $x_i \in \mathbb{R}^d$ is a feature vector, and $y_i$ is the corresponding label. We use $l(\hat{y}_i, y_i)$ to represent a loss function where $\hat{y}_i$ is the prediction of our model. We use $f_\theta(\cdot) : \mathbb{R}^d \to \mathbb{R}^o$ to denote the model parameterized by $\theta$ and use $\theta_0 \in \mathbb{R}^m$ to

denote random initialized parameters where $m$ is the number of model parameters. Bold font constant denotes a constant vector such as $\mathbf{0} = [0, 0, \cdots, 0]$ whose dimension depends on the corresponding situation. $\mathcal{D}$ denotes the underline data distribution, which is the same for training and testing.

Our theoretical results rely on an assumption that there exists a neighborhood $\Gamma_{\theta_0}$ for a random initialization point $\theta_0$ in which sample-wise optimization landscapes are almost convex and semi-smooth. Note that our assumption is weaker than that of (Allen-Zhu et al., 2019) since their analysis is focusing on the overall optimization landscape while we only consider the sample-wise optimization landscape which is a simpler case. We use $\{\theta_i^*\}_{i \in [n]} \in \Gamma_{\theta_0}$ to denote a local optima in the convex areas attached to the $i$-th sample near the initialization point $\theta_0$. Under this assumption, the overall optimization landscape is also convex and semi-smooth within the neighborhood $\Gamma_{\theta_0}$ as additive operations preserve both. We use $\theta^*$ to denote a local optimum within $\Gamma_{\theta_0}$ for the mini-batch optimization landscape. Note that $\theta^*$ always lie in the convex hull of $\{\theta_i^*\}_{i \in [n]}$. Second, we assume that only gradient-based optimizers [1] are used during training. Thus the optimizer eventually converges to $\theta^*$. Third, our theoretical analysis assumes that every Hessian in the set $\{\nabla^2 l(f_\theta(x_i), y_i) | \forall i \in [n], \theta \in \Gamma_{\theta_0}\}$ is almost diagonal as in the Neural Tangent Kernel (NTK) regime (Jacot et al., 2018). Section 5 shows that our method generalizes well to real-world networks that may violate this assumption.

**Sample-wise local optima density.** We use *sample-wise local optima density* to represent the relative closeness of $\{\theta_i^*\}_{i \in [n]}$. Given a dataset $\mathcal{S} = \{(x_i, y_i)\}_{i \in [n]}$, an objective function $l(\hat{y}_i, y_i)$, and a model class $f_\theta(\cdot)$, we use $\Psi_{\mathcal{S}, l}(f_{\theta_0}(\cdot))$ to measure the average distance between the local optima across samples $\{\theta_i^*\}_{i \in [n]}$ near a random initialization point $\theta_0$:

$$\Psi_{\mathcal{S}, l}(f_{\theta_0}(\cdot)) \quad = \quad \frac{\sqrt{\mathcal{H}}}{n^2} \sum_{i, j} \|\theta_i^* - \theta_j^*\|_1 \qquad (1)$$

$\mathcal{H} \in \mathbb{R}$ is a smoothness upper bound: $\forall k \in [m], i \in [n], [\nabla^2 l(f_\theta(x_i), y_i)]_{k,k} \leq \mathcal{H}$. This upper bound always exists due to the smoothness assumption. Intuitively, $\Psi_{\mathcal{S}, l}(f_{\theta_0}(\cdot))$ can be interpreted as the mean Manhattan distance with respect to each pair of $\{\theta_i^*\}_{i \in [n]}$ normalized by the inverse of the square root of the smoothness upper bound. The denser $\{\theta_i^*\}_{i \in [n]}$ are, the smaller $\Psi_{\mathcal{S}, l}(f_{\theta_0}(\cdot))$ is. In an ideal case, $\Psi_{\mathcal{S}, l}(f_{\theta_0}(\cdot)) = 0$ when all local optima are located at the same point.

### 3.3 MAIN RESULTS

We show the local optimum property of sample-wise optimization landscapes in the following lemma.

**Lemma 1** *There exists no saddle point in a sample-wise optimization landscape and every local optimum is a global optimum.*

Using Lemma 1, we can draw a relation between the training error $J = \frac{1}{n} \sum_i l(f_\theta(x_i), y_i)$ and $\Psi_{\mathcal{S}, l}(f_{\theta_0}(\cdot))$ using the following theorem.

**Theorem 2** *The training error of a network on a dataset $J = \frac{1}{n} \sum_i l(f_\theta(x_i), y_i)$ is upper bounded by $\frac{n^3}{2} \Psi_{\mathcal{S}, l}^2(f_{\theta_0}(\cdot))$, and the bound is tight when $\Psi_{\mathcal{S}, l}(f_{\theta_0}(\cdot)) = 0$.*

Finally, we show that $\Psi_{\mathcal{S}, l}(f_{\theta_0}(\cdot))$ also provides an upper bound for the generalization performance of a network measured by population loss.

**Theorem 3** *Given that $Var_{(x_u, y_u) \sim \mathcal{D}}[\|\theta^* - \theta_u^*\|_1^2]$ is bounded by $\sigma^2$ where $\theta_u^*$ is a local optimum attached to the convex area near $\theta_0$ for $l(f_\theta(x_u), y_u)$. With probability $1 - \delta$, the true population loss is upper bounded by $\frac{n^3}{2} \Psi_{\mathcal{S}, l}^2(f_{\theta_0}(\cdot)) + \frac{\sigma}{\sqrt{n\delta}}$.*

A formal proof of all theoretical results is available in Appendix A.2.

**Main takeaways:** A key takeaway of our theoretical results is that $\Psi_{\mathcal{S}, l}(f_{\theta_0}(\cdot))$ closely relates to an upper bound of the training and generalization performance of a network. Albeit theoretically sound, $\Psi_{\mathcal{S}l}(f_{\theta_0}(\cdot))$ is intractable to be directly measured. Instead, we derive a simple yet accurate metric to reflect $\Psi_{\mathcal{S}, l}(f_{\theta_0}(\cdot))$, which we present in the next section.

---

[1]Gradient descent with infinitesimal step size

## 4 GRADSIGN

Inspired by the theoretical results derived above, we introduce GradSign, a simple yet accurate metric for model performance inference. The key idea behind GradSign is a quantity to statistically reflect the relative value of $\Psi_{\mathcal{S},l}(f_{\theta_0}(\cdot))$. Specifically,

$$\Psi_{\mathcal{S},l}(f_{\theta_0}(\cdot)) \propto C - \sum_k \sum_{i,j} P(\text{sign}([\nabla_\theta l(f_\theta(x_i), y_i)|_{\theta_0}]_k) = \text{sign}([\nabla_\theta l(f_\theta(x_j), y_j)|_{\theta_0}]_k)) \quad (2)$$

where $C$ is a constant and $k$ is the vector index. Detailed derivation is given in Appendix A.2. Using the above relation, we can infer $\Psi_{\mathcal{S},l}(f_{\theta_0}(\cdot))$ by directly measuring the signs of the gradients for a mini-batch of training samples at a randomly initialized point instead of going through an end-to-end training process.

To enable more efficient calculation of $\Psi_{\mathcal{S},l}(f_{\theta_0}(\cdot))$, we make a further simplification and use the following sample observation:

$$\sum_k |\sum_i \text{sign}([\nabla_\theta l(f_\theta(x_i), y_i)|_{\theta_0}]_k)| \quad (3)$$

to infer the true probability $\sum_k \sum_{i,j} P(\text{sign}([\nabla_\theta l(f_\theta(x_i), y_i)|_{\theta_0}]_k) = \text{sign}([\nabla_\theta l(f_\theta(x_j), y_j)|_{\theta_0}]_k))$ whose relation is given by:

$$\frac{1}{n^2} \sum_{i,j} \mathbb{1}_{\text{sign}([\nabla_\theta l(f_\theta(x_i), y_i)|_{\theta_0}]_k) = \text{sign}([\nabla_\theta l(f_\theta(x_j), y_j)|_{\theta_0}]_k)} \quad (4)$$

$$\propto |\sum_i \text{sign}([\nabla_\theta l(f_\theta(x_i), y_i)|_{\theta_0}]_k)|^2 \quad (5)$$

The proof of this simplification is included in Appendix A.2. Given the above relationship, we formally state our algorithm pipeline in Algorithm 1. Note that a higher GradSign score indicates better model performance as we have an inverse correlation in Equation (2).

---

**Algorithm 1:** GradSign

---

**Result:** GradSign score $\tau_f$ for a function class $f_\theta$
Given $\mathcal{S} = \{(x_i, y_i)\}_{i \in [n]}$, randomly select initialization point $\theta_0$;
Initialize $g[n, m]$;
**for** $i = 1, 2, \cdots, n$ **do**
    **for** $k = 1, 2, \cdots, m$ **do**
        | $g[i, k] = \text{sign}([\nabla_\theta l(f_\theta(x_i), y_i)|_{\theta_0}]_k)$
    **end**
**end**
$\tau_f = \sum_k |\sum_i g[i, k]|$;
**return** $\tau_f$

---

## 5 EXPERIMENTS

In this section, we empirically verify the effectiveness of our metric against existing gradient-based methods on three neural architecture search (NAS) benchmarks, including NAS-Bench-101 (Ying et al., 2019), NAS-Bench-201 (Dong & Yang, 2020) and NDS (Radosavovic et al., 2019)[2]. Theory-based, sample-based, and learning-based methods are excluded in our evaluation, as they either require further training processes or have strong assumptions not suitable for generic architectures.

**Baselines.** We compare GradSign against existing gradient-based methods, including snip (Lee et al., 2018), grasp (Wang et al., 2020), fisher (Turner et al., 2019), and Synflow. In addition, we also include grad_norm as a heuristic method and a one-shot MPI metric NASWOT(Mellor et al., 2021). Since all gradient-based methods share a similar calculation pipeline (i.e., evaluating the gradients of a mini-batch at a random initialization point), we set the initialization mode and batch size to be the same across all methods to guarantee fairness. Experimental setup details are included in Appendix A.3. To align with the experimental setup of prior work (Abdelfattah et al., 2021b;

---

[2]All datasets have consented for research purposes and no identifiable personal information is included.

Mellor et al., 2021), we use two criteria to evaluate the correlations between different metrics and test accuracies across approximately 20k networks:

**Spearman's** $\rho$ (Daniel et al., 1990) characterizes the monotonic relationships between two variables. The correlation score is restricted in range [-1, 1], where $\rho = 1$ denotes a perfect positive monotonic relationship and $\rho = -1$ denotes a perfect negative monotonic relationship. Following prior work, we use Spearman's $\rho$ to evaluate gradient-based methods on NAS-Bench-101 and NAS-Bench-201.

**Kendall's Tau**. Similar to Spearman's $\rho$ as a correlation measurement, Kendall's Tau is also restricted between [-1, 1]. While Spearman's $\rho$ is more sensitive to error and discrepancies, Kendall's Tau is more robust with a smaller gross error sensitivity. We use Kendall's Tau to quantify the correlation between one-shot metric scores and model testing accuracies over the NDS search space.

## 5.1 NAS-BENCH-101

NAS-Bench-101 is the first dataset targeting large-scale neural architecture space, containing 423k unique convolutional architectures trained on the CIFAR-10 dataset. The benchmark provides the test accuracy of each architecture in the search space, which we use to calculate the corresponding Spearman's $\rho$. We use a randomly sampled subset with approximately $4500$ architectures of the original search space and a batch size of 64 in this experiment. Table 2 summarizes the results. GradSign significantly outperforms existing gradient-based methods and heuristic approaches and improves the Spearman's $\rho$ score by $25\%$ compared to the best existing method ($0.363 \rightarrow 0.449$).

Table 2: Performance of existed MPI methods (gradient-based + NASWOT) on NAS-Bench-101 evaluated by Spearman's $\rho$.

| Dataset | grad_norm | snip | grasp | fisher | Synflow | NASWOT | GradSign |
|---------|-----------|------|-------|--------|---------|--------|----------|
| CIFAR10 | 0.263 | 0.189 | 0.315 | 0.3 | 0.363 | 0.324 | **0.449** |

## 5.2 NAS-BENCH-201

NAS-Bench-201 is an extended version of NAS-Bench-101 with a different search space, containing 15,625 cell-based candidate architectures evaluated across three datasets: CIFAR-10, CIFAR-100 (Krizhevsky, 2009) and ImageNet 16-120 (Russakovsky et al., 2015). The benchmark provides the test accuracies on the three datasets for all candidate architectures in the search space. We evaluate Spearman's $\rho$ scores for GradSign and existing gradient-based methods. The experiments were conducted overall 15,265 architectures in NAS-Bench-201. The batch size is set to 64. The results on the three datasets are summarized in Table 3. GradSign consistently achieves the best performance across all three datasets and improves the Spearman's $\rho$ scores by $\approx 4\%$ over the best existing approaches. This improvement is significant as the more Spearman's $\rho$ approaches $1$, the more difficult it can be further improved.

Table 3: Performance of existed MPI methods (gradient-based + NASWOT + ZenNAS) on NAS-Bench-201 evaluated by Spearman's $\rho$.

| Dataset | ZenNAS | grad_norm | snip | grasp | fisher | Synflow | NASWOT | GradSign |
|---------|--------|-----------|------|-------|--------|---------|--------|----------|
| CIFAR10 | -0.016 | 0.594 | 0.595 | 0.51 | 0.36 | 0.737 | 0.728 | **0.765** |
| CIFAR100 | -0.041 | 0.637 | 0.637 | 0.549 | 0.386 | 0.763 | 0.703 | **0.793** |
| ImageNet16-120 | 0.032 | 0.579 | 0.579 | 0.552 | 0.328 | 0.751 | 0.696 | **0.783** |

We further select 1000 architecture candidates randomly in the NAS-Bench-201 search space and visualize their testing accuracies against GradSign scores in Figure 2. Figure 2 shows a highly positive correlation between the GradSign score and actual test accuracy of 1000 architectures. A higher GradSign score indicates higher confidence for the statistical performance of architecture. Note that the GradSign scores show a clustering pattern, which may correspond to different architecture classes in the NAS-Bench-201 search space.

## 5.3 NAS DESIGN SPACE (NDS)

NDS is a unified searching framework that includes five different design spaces: NAS-Net (Zoph et al., 2018), AmoebaNet (Real et al., 2019), PNAS (Liu et al., 2018a), ENAS (Pham et al., 2018),

DARTS (Liu et al., 2018b). Each space contains approximately one thousand networks fully trained on the CIFAR-10 dataset. We include the performance of our method along with *grad_norm*, Synflow, and NASWOT on all five design spaces evaluated by Kendall's Tau and show the results in Table 4. GradSign significantly and consistently outperforms all other MPI methods in all five design spaces.

Table 4: Performance of existed MPI methods on five design spaces in NDS trained over CIFAR-10 evaluated by Kendall's Tau.

|  | DARTS | ENAS | PNAS | NASNet | Amoeba |
|---|---|---|---|---|---|
| *grad_norm* | 0.28 | -0.02 | -0.01 | -0.08 | -0.10 |
| Synflow | 0.37 | 0.02 | 0.03 | -0.03 | -0.06 |
| NASWOT | 0.48 | 0.34 | 0.31 | **0.31** | 0.20 |
| GradSign | **0.54** | **0.43** | **0.40** | **0.31** | **0.24** |

## 5.4 ARCHITECTURE SELECTION

We evaluate whether GradSign can be directly used to select highly accurate architectures in a NAS search space. To pick a top architecture, we randomly sample $N$ candidates in a NAS search space, choose the one with the highest GradSign score, and measure its validation/test accuracies (mean±std). We compare GradSign with Synflow, NASWOT, Random, and Optimal, where Random uniformly samples architectures in the search space, while Optimal always chooses the best architecture across $N$ candidates. The results[3] are summarized in Table 5. $N$ in parenthesis indicates the number of architectures sampled in each run. All methods can generally find more accurate architectures with a high $N$. In addition to outperforming Synflow and NASWOT, GradSign ($N = 100$) can also find better networks even compared to NASWOT($N = 1000$). The results show that GradSign can directly identify accurate architectures besides highly correlating to networks' test accuracies.

Table 5: Mean ± std accuracy evaluated on NAS-Bench-201. All results are averaged over 500 runs. All searches are conducted on CIFAR-10 while the selected architectures are evaluated on CIFAR-10, CIFAR-100, and ImageNet16-120. $N$ in parenthesis is the number of networks sampled in each run.

| Methods | CIFAR-10 | | CIFAR-100 | | ImageNet16-120 | |
|---|---|---|---|---|---|---|
| | Validation | Test | Validation | Test | Validation | Test |
| Synflow(N=100) | **89.83±0.75** | 93.12±0.52 | 69.89±1.87 | 69.94±1.88 | 41.94±4.13 | 42.26±4.26 |
| NASWOT(N=100) | 89.55±0.89 | 92.81±0.99 | 69.35±1.70 | 69.48±1.70 | 42.81±3.05 | 43.10±3.16 |
| NASWOT(N=1000) | 89.69±0.73 | 92.96±0.81 | 69.98±1.22 | 69.86±1.21 | **44.44±2.10** | **43.95±2.05** |
| GradSign(N=100) | **89.84±0.61** | **93.31±0.47** | **70.22±1.32** | **70.33±1.28** | 42.07±2.78 | 42.42±2.81 |
| Random | 83.20±13.28 | 86.61±13.46 | 60.70±12.55 | 60.83±12.58 | 33.34±9.39 | 33.13±9.66 |
| Optimal(N=100) | 91.05±0.28 | 93.84±0.23 | 71.45±0.79 | 71.56±0.78 | 45.37±0.61 | 45.67±0.64 |

## 5.5 GRADSIGN-ASSISTED NEURAL ARCHITECTURE SEARCH

Besides evaluating GradSign on the Spearman's $\rho$ and Kendall's Tau scores as prior work, we also integrate GradSign into various neural architecture search algorithms and evaluate how GradSign can assist neural architecture search on real-world tasks. Specifically, we integrate GradSign into four NAS algorithms: REA, REINFORCE, BOHB and RS. We design a corresponding method for each NAS algorithm that uses the GradSign scores of candidate architectures to guide the search. Specifically, we integrate GradSign into each NAS algorithm by replacing the random selection of architectures with GradSign-assisted selection. We name these GradSign-assisted variants G-REA, G-REINFROCE, G-HB, and G-RS, and describe their algorithm details in Appendix A.

To evaluate how GradSign can improve the search procedure of NAS algorithms, we run each algorithm with and without GradSign's assistance for 500 runs on NAS-Bench-201 and report the validation and test accuracies of the best-discovered architecture in each run. Following prior work (Mellor et al., 2021; Dong & Yang, 2020), all searches are conducted on the CIFAR-10 dataset with a time budget of 12000s while the performance is evaluated on CIFAR-10, CIFAR-100 and

---

[3]the results for NASWOT are referenced from their paper (Mellor et al., 2021)

ImageNet16-120. The baselines also include A-REA (Mellor et al., 2021), a variant of REA that uses the NASWOT scores at the initial population selection phase.

Table 6 shows the results. The GradSign-assisted NAS algorithms outperform their counterparts by improving test accuracy by up to 0.3%, 1.1%, and 1.0% on the three datasets.

Table 6: Mean $\pm$ std accuracy evaluated over NAS-Bench-201. All results are averaged over 500 runs. To make a fair comparison across all the methods, the search is performed on CIFAR-10 dataset while the architectures' performance are evaluated over CIFAR-10, CIFAR-100 and ImageNet16-120. All the methods have a search time budget of 12000s. Note that the benchmark results might not match with the original paper as we have run all the experiments from start in a environment different from Dong & Yang (2020).

| Methods | CIFAR-10 | | CIFAR-100 | | ImageNet16-120 | |
|---|---|---|---|---|---|---|
| | Validation | Test | Validation | Test | Validation | Test |
| REA | 91.08±0.45 | 93.85±0.44 | 71.59±1.33 | 71.64±1.25 | 44.90±1.20 | 45.25±1.41 |
| A-REA | 91.20±0.27 | - | 71.95±0.99 | - | **45.70±1.05** | - |
| G-REA | **91.27±0.58** | **94.10±0.52** | **72.64±1.57** | **72.70±1.50** | 45.69±1.33 | **45.7±1.32** |
| RS | 90.93±0.37 | 93.72±0.38 | 70.96±1.12 | 71.07±1.07 | 44.47±1.08 | 44.61±1.22 |
| G-RS | **91.24±0.21** | **94.02±0.21** | **72.15±0.77** | **72.20±0.76** | **45.38±0.79** | **45.77±0.79** |
| REINFORCE | 90.32±0.89 | 93.21±0.82 | **70.03±1.75** | 70.14±1.73 | 43.57±2.09 | 43.64±2.24 |
| G-REINFORCE | **90.47±0.55** | **93.37±0.47** | 70.00±1.20 | **70.20±1.29** | **44.33±1.25** | **44.05±1.48** |
| BOHB | 90.84±0.49 | 93.64±0.49 | 70.82±1.29 | 70.92±1.26 | 44.36±1.37 | 44.50±1.50 |
| G-HB | **91.18±0.26** | **93.96±0.25** | **71.92±0.92** | **71.99±0.85** | **45.29±0.84** | **45.53±0.92** |

## 6 CONCLUSION

In this paper, we propose a model performance inference metric GradSign and provide theoretical foundations to support our metric. Instead of focusing on full batch optimization landscape analysis, we move a step further to sample-wise optimization landscape properties, which give us additional information to uncover the quality of the local optima encountered on the optimization trajectory. We propose $\Psi_{\mathcal{S},l}(f_{\theta_0}(\cdot))$ to quantitatively characterize the potential of a model $f_\theta(\cdot)$ at a random initialization point $\theta_0$ based on our theory results. Finally, we design the GradSign metric to statistically infer the value of $\Psi_{\mathcal{S},l}(f_{\theta_0}(\cdot))$ to give out our final score for model performance inference. Empirically, we have demonstrated that our method consistently achieves the best correlation with true model performance among all other gradient-based metrics. In addition, we also verified the practical value of our method in assisting existed NAS algorithms to achieve better results. Given that our metric is generic and promising, we believe that our work not only assists in accelerating MPI-related applications but sheds some light on optimization landscape analysis as well. Meanwhile, the effectiveness of our method may further reduce the energy cost introduced by modern NAS algorithms. In addition, one of the future work of GradSign can be adding normalization across different architecture classes to tackle the clustering problem in Figure 2.

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

# A  APPENDIX

## A.1  FIGURE

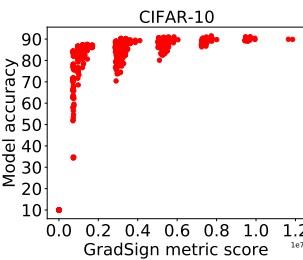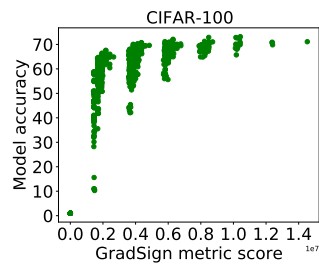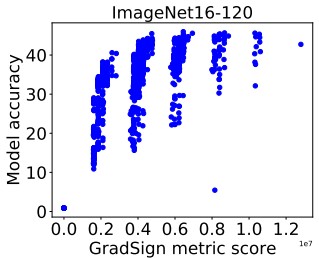

Figure 2: Visualization of model testing accuracy versus GradSign metric score on CIFAR10, CIFAR100, ImageNet16-120.

## A.2  PROOF

**Lemma 1 Proof:**   For a single training sample $(x_i, y_i)$, we minimize its objective function $l(f_\theta(x_i), y_i)$ with gradient descent:

$$\nabla_\theta l(f_\theta(x_i), y_i) \quad = \quad \frac{\partial l(f_\theta(x_i), y_i)}{\partial f_\theta(x_i)} \cdot f'_\theta(x_i) \tag{6}$$

At any local optimal point $\theta_i^*$, we have $\nabla_\theta l(f_\theta(x_i), y_i)|_{\theta_i^*} = \mathbf{0}$, but $f'_\theta(x_i) \neq \mathbf{0}$ for conventional neural architectures with at least one dense layer [4]. Therefore, we show $\frac{\partial l(f_\theta(x_i), y_i)}{\partial f_\theta(x_i)}|_{\theta_i^*}$ must be equal to $0$. For the commonly used objective functions, such as Mean Squared Error Loss and Cross Entropy Loss, this derivative is equal to $C(f_\theta(x_i) - y_i)$, where $C$ is a non-zero constant. Hence we have $f_{\theta_i^*}(x_i) = y_i$ and $l(f_{\theta_i^*}(x_i), y_i) = 0$ at local optima $\theta_i^*$, which makes $\theta_i^*$ also a global optima as it is impossible to obtain a lower loss value for this single sample. In addition, at local optima $\theta_i^*$, we have:

$$\nabla_\theta^2 l(f_\theta(x_i), y_i) \quad = \quad \nabla_\theta \left( \frac{\partial l(f_\theta(x_i), y_i)}{\partial f_\theta(x_i)} \cdot f'_\theta(x_i) \right) \tag{7}$$

$$= \quad \frac{\partial^2 l(f_\theta(x_i), y_i)}{\partial f_\theta(x_i)^2} \cdot f'_\theta(x_i) f'_\theta(x_i)^\top + \frac{\partial l(f_\theta(x_i), y_i)}{\partial f_\theta(x_i)} \cdot f''_\theta(x_i) \tag{8}$$

$$= \quad C \cdot f'_\theta(x_i) f'_\theta(x_i)^\top \tag{9}$$

The above equation implies that $\nabla_\theta^2 l(f_\theta(x_i), y_i)$ is a positive semi-definite matrix, since $C > 0$. This concludes the proof of non existence of saddle points in a sample-wise optimization landscape. This result aligns with our convexity assumptions in Section 3.2.

**Theorem 2 Proof:**   Recall that $\theta^*$ denotes a local optima of $J$ which could be reached by a gradient-flow based optimizer start from $\theta_0$. Since $\theta_0$ is randomly sampled and $[\nabla^2 l(f_\theta(x_i), y_i)]_{k,k} \leq \mathcal{H}$, we have:

$$J \quad = \quad \frac{1}{n} \sum_i l(f_{\theta^*}(x_i), y_i) \tag{10}$$

$$\leq \quad \frac{1}{n} \sum_i \mathcal{H} \cdot \|\theta^* - \theta_i^*\|_2^2 \tag{11}$$

$$\leq \quad \frac{\mathcal{H}}{n} \sum_i \|\theta^* - \theta_i^*\|_1^2 \tag{12}$$

$$\leq \quad \frac{\mathcal{H}}{n} \sum_{i,j} \|\theta_i^* - \theta_j^*\|_1^2 \tag{13}$$

$$\leq \quad n^3 \Psi_{\mathcal{S},l}^2(f_{\theta_0}(\cdot)) \tag{14}$$

---

[4] The gradient values corresponding to the bias term of the last dense layer are always non-zero.

where Eq 10 $\rightarrow$ Eq 11 uses the basic property of the smoothness upper bound $\mathcal{H}$ and the fact that each local optimum $\theta_i^*$ satisfies $l(f_{\theta_i^*}(x_i), y_i) = 0$. Eq 11 $\rightarrow$ Eq 12 uses Jensen Inequality for square root operators. Eq 12 $\rightarrow$ Eq 13 is derived from the fact that for each dimension in $(\theta^* - \theta_i^*)$ we have:

$$|[\theta^* - \theta_i^*]_k| \leq \sum_j |[\theta_j^* - \theta_i^*]_k| \tag{15}$$

Otherwise, $\theta^*$ dose not lie in the convex hull of $\{\theta_i^*\}_{i \in [n]}$ which contradicts with our assumption stated in Section 3.2. The bound is tight when $\theta_i^* = \theta_j^*, \forall i, j \in [n]$.

Eq 10$\rightarrow$Eq 11: As we have $\nabla^2 l(f_\theta(x_i), y_i) \preceq \mathcal{H}\mathbb{I}$ and $l(f_{\theta_i^*}(x_i), y_i) = 0, \nabla_\theta l(f_\theta(x_i), y_i)|_{\theta_i^*} = \mathbf{0}$, we could derive the following inequality:

$$l(f_{\theta^*}(x_i), y_i) \leq l(f_{\theta_i^*}(x_i), y_i) + \nabla_\theta l(f_{\theta_i^*}(x_i), y_i)^\top (\theta^* - \theta_i^*) + \frac{\mathcal{H}}{2}\|\theta^* - \theta_i^*\|_2^2 \tag{16}$$

$$= \frac{\mathcal{H}}{2}\|\theta^* - \theta_i^*\|_2^2 \tag{17}$$

we thus have $\frac{1}{n}\sum_i l(f_{\theta^*}(x_i), y_i) \leq \frac{1}{2n}\sum_i \mathcal{H} \cdot \|\theta^* - \theta_i^*\|_2^2$.

**Theorem 3 Proof:** Let $\mathbb{E}_{(x_u, y_u) \sim \mathcal{D}}[l(f_\theta^*(x_u), y_u)]$ denotes the true population error. With probability $1 - \delta$, we have:

$$\mathbb{E}_{(x_u, y_u) \sim \mathcal{D}}[l(f_\theta^*(x_u), y_u)] \leq \mathcal{H}\mathbb{E}_{(x_u, y_u) \sim \mathcal{D}}[\|\theta^* - \theta_u^*\|_1^2] \tag{18}$$

$$\leq \frac{\mathcal{H}}{n}\sum_{i=1}^n \|\theta^* - \theta_i^*\|_1^2 + \frac{\sigma}{\sqrt{n\delta}} \tag{19}$$

$$\leq n^3 \Psi_{\mathcal{S},l}^2(f_{\theta_0}(\cdot)) + \frac{\sigma}{\sqrt{n\delta}} \tag{20}$$

where $n$ and $\sigma$ are constants, $\mathcal{S}$ is a training dataset, and $\mathcal{D}$ denotes its underlying data distribution. This implies that $\Psi_{\mathcal{S},l}(f_{\theta_0}(\cdot))$ is an accurate indicator for the true population loss. Eq 18 $\rightarrow$ Eq 19 uses Chebyshev's inequality, while Eq 19 $\rightarrow$ Eq 20 uses the same inequality derived in **Claim 2**.

**Algorithm Proof:** Given that sample-wise local optima $\{[\theta_i^*], i \in [n]\}$ are contained in the convex area around $\theta_0$. We derive the following property:

$$\text{sign}([\theta_i^* - \theta_0]_k) = \text{sign}([\nabla_\theta l(f_\theta(x_i), y_i)|_{\theta_0}]_k) \tag{21}$$

Since $\theta_0$ is a randomly chosen initialization point, without loss of generality, we assume $\theta_0$ is sampled from a hypercube $[-a, a]$. Thus we have:

$$P(\text{sign}([\nabla_\theta l(f_\theta(x_i), y_i)|_{\theta_0}]_k) \neq \text{sign}([\nabla_\theta l(f_\theta(x_j), y_j)|_{\theta_0}]_k)) = \frac{|[\theta_i^*]_k - [\theta_j^*]_k|}{2a} \tag{22}$$

Where $P(\text{sign}([\nabla_\theta l(f_\theta(x_i), y_i)|_{\theta_0}]_k) \neq \text{sign}([\nabla_\theta l(f_\theta(x_j), y_j)|_{\theta_0}]_k))$ denotes the probability for $[\nabla_\theta l(f_\theta(x_i), y_i)|_{\theta_0}]_k$ and $[\nabla_\theta l(f_\theta(x_j), y_j)|_{\theta_0}]_k$ having different signs. Notice that we have completely dropped the dependency for $\theta_i^*$ at this point and can simply infer from $\text{sign}([\nabla_\theta l(f_\theta(x_i), y_i)|_{\theta_0}]_k)$. To complete our proof:

$$\Psi_{\mathcal{S},l}(f_{\theta_0}(\cdot)) = \frac{\sqrt{\mathcal{H}}}{n}\sum_{i,j} \|\theta_i^* - \theta_j^*\|_1 \tag{23}$$

$$= \frac{2a\sqrt{\mathcal{H}}}{n}\sum_k \sum_{i,j} P(\text{sign}([\nabla_\theta l(f_\theta(x_i), y_i)|_{\theta_0}]_k) \neq \text{sign}([\nabla_\theta l(f_\theta(x_j), y_j)|_{\theta_0}]_k)) \tag{24}$$

$$\propto n^2 - \sum_k \sum_{i,j} P(\text{sign}([\nabla_\theta l(f_\theta(x_i), y_i)|_{\theta_0}]_k) = \text{sign}([\nabla_\theta l(f_\theta(x_j), y_j)|_{\theta_0}]_k)) \tag{25}$$

**Simplification Proof::** for each $i, j$ and a given $k$, we could estimate $P(\text{sign}([\nabla_\theta l(f_\theta(x_i), y_i)|_{\theta_0}]_k) = \text{sign}([\nabla_\theta l(f_\theta(x_j), y_j)|_{\theta_0}]_k))$ using:

$$\frac{|\text{sign}([\nabla_\theta l(f_\theta(x_i), y_i)|_{\theta_0}]_k) + \text{sign}([\nabla_\theta l(f_\theta(x_j), y_j)|_{\theta_0}]_k)|}{2} \tag{26}$$

which is valid (not equal to zero) only when $\text{sign}([\nabla_\theta l(f_\theta(x_i), y_i)|_{\theta_0}]_k) ==$ $\text{sign}([\nabla_\theta l(f_\theta(x_j), y_j)|_{\theta_0}]_k)$. Suppose we have p positive and n-p negative $\text{sign}([\nabla_\theta l(f_\theta(x_i), y_i)|_{\theta_0}]_k)$ for n samples, since we only care about samples share the same sign, the original probability estimation is $\frac{(n-p)^2 + p^2}{n^2} = \frac{1}{2} + \frac{(n-2p)^2}{2n^2}$. Thus we only need to measure the quantity $|n - 2p|$ which simply equals to $|\sum_i \text{sign}([\nabla_\theta l(f_\theta(x_i), y_i)|_{\theta_0}]_k)|$.

### A.3 EXPERIMENTS SETUP

The code we used during experimentation was mainly based on existed code base Abdelfattah et al. (2021b); Mellor et al. (2021)Abdelfattah et al. (2021b) which is under Apache-2.0 License.

The hardwares we used were Amazon EC2 C5 instances with no GPU involved and p3 instance with one V100 Tensor Core GPU.

As our methods is gradient-based which is training free, we don't need to split our dataset. For Spearman's correlation measurement on NAS-Bench-201, we set batch size to 64, which is used by most baselines. For the Kendall's Tau experiment, other accuracy comparison experience and the GradSign assisted algorithms, we used a batch size of 128, also to match the batch size in other baselines (NASWOT). We use Pytorch default parameter initialization for all architectures. Random seed in correlation experiments is set to 42 which is also randomly chosen. For accuracy experiments, our results are summarized over 500 runs whose random seed are chosen randomly for each run. For the correlation evaluation of each individual architecture, we only use one $\theta_0$ for minimizing computational cost. Our approach can be easily generalized to an average of multiple $\theta_0$s and can trade-off between efficiency and accuracy.

### A.4 ADDITIONAL RESULTS

**Sample-based:** Fig 3 compares EconNAS, a sample-based method, with existing gradient-based methods on MPI. Results of EconNAS are referenced from Abdelfattah et al. (2021b). To achieve a similar MPI performance as GradSign, EconNAS needs 500 minibatches of samples for each candidate's proxy training, while all gradient-based methods (including GradSign) require only one minibatch. By increasing the number of minibatches, EconNAS can achieve higher Spearman's $\rho$ scores, which eventually converge to 0.85. At that point overfitting takes place and the score cannot be further improved.

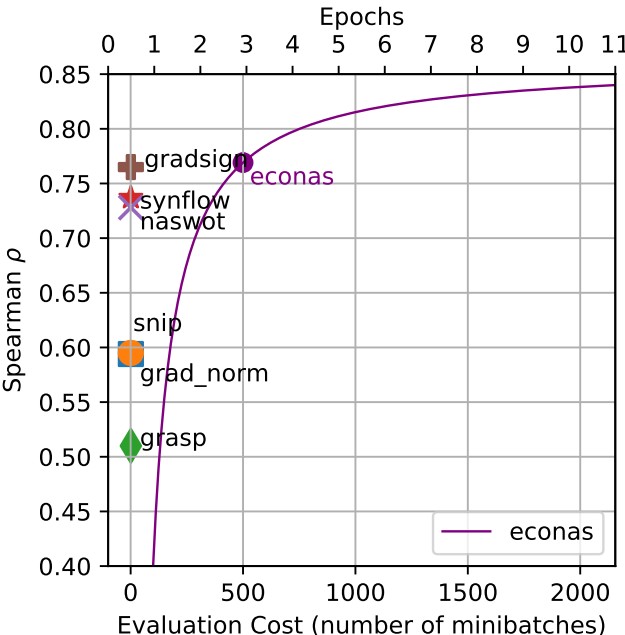

Figure 3: Comparison with sample-based methods (EconNAS) on NAS-Bench-201 across CIFAR-10. EconNAS requires more than 500 minibatches to have a better performance than GradSign while gradient-based methods only require 1 minibatch.

**Learning-based:** Table 7 compares GradSign and existing learning-based methods (MLP, LSTM, and GATES) on the Kendall's Tau correlation score. The MLP, LSTM, and GATES results are referenced from Ning et al. (2020). For MLP and LSTM (Wang et al., 2019), the predictor uses Multi Layer Perceptron (MLP) and Long Short Term Memory (LSTM) as the base predictor, while GATES (Ning et al., 2020) uses Graph Neural Network (GNN) as the base predictor. All results are obtained on NAS-Bench-201 and the GradSign's score is averaged over all three datasets (CIFAR-10, CIFAR-100 and ImageNet16-120) as (Ning et al., 2020) does not provide which dataset is used for calculating Kendall's Tau.

To achieve a similar score as GradSign, MLP, LSTM and GATES-1 require an average of 1959, 978 and 1959 minibatches per sample respectively to prepare the dataset for training the predictors. Although GATES-2 achieves a better correlation score than GradSign, it still needs 195 minibatches per sample to prepare the dataset for training the GATES-2 predictor. In addition to the cost of preparing a training dataset, each predictor also has to be trained on the dataset as well, which involves 200 more epochs while the cost of evaluating GradSign is one mini-batch. With less mini-batches evaluated for learning-based methods, their training set sizes shrink significantly (e.g., 195 mini-batches equal to 78 training samples and 7813 testing samples). This may result in overfitting to the training set.

Table 7: Comparison with learning-based methods (MLP, LSTM and GATES) on NAS-Bench-201. GATES-1 represents GATES predictor with only one layer and GATES-2 denotes GATES predictors with more than one layers.

|  | Kendall's Tau | Average minibatches per sample |
|---|---|---|
| MLP | 0.5388 | 1959 |
| LSTM | 0.6407 | 978 |
| GATES-1 | 0.45 | 1959 |
| GATES-2 | 0.7401 | 195 |
| GradSign | 0.6016 | 1 |

We also includes the evaluation of GradSign for both MPI correlation performance and GradSign-assisted NAS algorithms on the latest version of NAS-Bench-201 (NATS-Bench) across three datasets (CIFAR-10, CIFAR-100 and ImageNet16-120) in Table 8, Table 9. Results show GradSign is robust against hyper-parameter tuning as long as the trained networks can converge to near optimal. Also notice that GradSign-assisted NAS algorithms could not only achieve a better accuracy but lower variance as well compared to their non-assisted counterparts.

Table 8: Spearman's $\rho$ evaluated on the latest version of NAS-Bench-201 (NATS-Bench)

|  | CIFAR-10 | CIFAR-100 | ImageNet16-120 |
|---|---|---|---|
| NAS-Bench-201 | 0.765 | 0.793 | 0.783 |
| NATS-Bench | 0.760 | 0.792 | 0.784 |

Table 9: Mean $\pm$ std accuracy evaluated over NATS-Bench. All results are averaged over 500 runs. To make a fair comparison across all the methods, the search is performed on CIFAR-100 dataset while the architectures' performance are evaluated over CIFAR-10, CIFAR-100 and ImageNet16-120. All the methods have a search time budget of 12000s. Note that the benchmark results might not match with the original paper as we have run all the experiments from start in a environment different from Dong & Yang (2020).

| Methods | CIFAR-10 | | CIFAR-100 | | ImageNet16-120 | |
|---|---|---|---|---|---|---|
| | Validation | Test | Validation | Test | Validation | Test |
| REA | 91.06±0.49 | 93.84±0.45 | 71.53±1.31 | 71.60±1.27 | 44.82±1.23 | 45.18±1.37 |
| G-REA | **91.35±0.35** | **94.15±0.32** | **72.67±1.05** | **72.65±0.97** | **45.55±0.96** | **45.99±0.93** |
| RS | 90.95±0.28 | 93.77±0.26 | 71.01±0.97 | 71.15±0.95 | 44.58±0.95 | 44.73±1.10 |
| G-RS | **91.23±0.22** | **94.02±0.22** | **72.12±0.82** | **72.15±0.78** | **45.43±0.74** | **45.83±0.80** |
| REINFORCE | 90.92±0.38 | 93.71±0.37 | 71.04±1.02 | 71.17±1.12 | 44.56±0.97 | 44.80±1.18 |
| G-REINFORCE | **91.20±0.23** | **93.98±0.23** | **71.93±0.91** | **72.05±0.89** | **45.28±0.77** | **45.64±0.86** |

To demonstrate the potential for GradSign in a more complicated computer vision task, we compare the performance of GradSign with ZenNAS following their setups and search space. Due to the limitation of computational resources[5], we only run 10000 evolution iterations using solely Zen score or GradSign score to select the architecture candidate and 20 epochs to train the selected architecture. Following ZenNAS's setup, EfficientNet-B3 is used as teacher network when training selected architectures. Though the top-1 validation accuracy of GradSign in the first 20 epochs is slightly better than Zen, we should note that this process can highly depend on the random seed for evolution search phase. As we mentioned before, ZenNAS uses linear region analysis which makes it less flexible for arbitrary activation functions. On the other hand, since ZenNAS calculates an architecture complexity related score which is both dataset independent and initialization independent, it can be too general and results in low Spearman's $\rho$ as shown in Table 3.

---

[5]the original setup in ZenNAS could take up to 8 months for training 480 epochs on ImageNet-1k

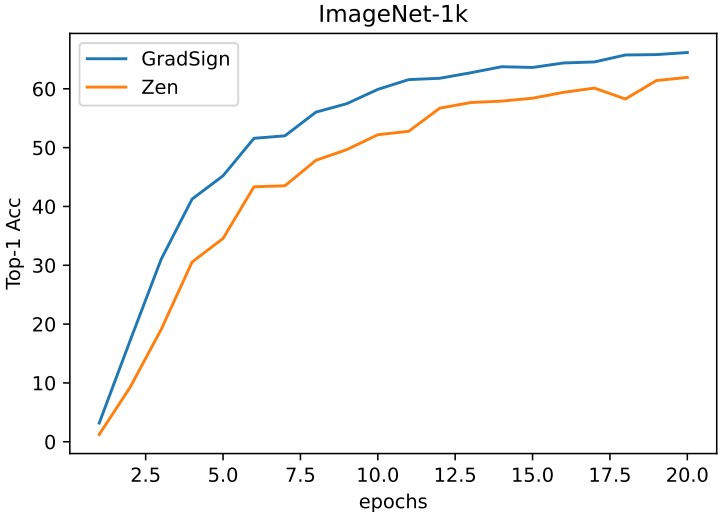

Figure 4: Comparison with ZenNAS in their search space on ImageNet-1k. Due to the limitation of computational resources, we only run 10000 evolution iterations and 20 epochs to train the selected architecture. We plot the top-1 prediction accuracy along training for two methods (ZenNAS, GradSign).

## A.5   GRADSIGN ASSISTED NAS ALGORITHMS

---

**Algorithm 2:** G-REA

---

**Result:** Find the best performing architecture given the time constraint for 12000s
Population = [];
History = [];
population_size;
sample_size;
pool_size;
**for** $1, 2, \cdots$ ,*population_size* **do**
    model = random.arch();
    model.acc, model.time_cost = eval(model);
    Population.append(model);
    History.append(model);
**end**
**while** *not exceeding time budget* **do**
    Sample = [];
    **for** $1, 2, \cdots$ ,*sample_size* **do**
        Sample.append(random.choice(Population))
    **end**
    parent = max_acc(Sample);
    GradSign_pool = [];
    **for** $1, 2, \cdots$ ,*pool_size* **do**                  `/* GradSign assisted part */`
        model = mutate_arch(parent);
        model.score = GradSign(model);
        GradSign_pool.append(model);
    **end**
    child = max_score(GradSign_pool);
    Population.append(child);
    History.append(child);
    Population.popleft();
**end**
**return** max_acc(History)

---

**Algorithm 3:** G-RS

---

**Result:** Find the best performing architecture given the time constraint for 12000s
History = [];
pool_size;
**while** *not exceeding time budget* **do**
    GradSign_pool = [];
    **for** $1, 2, \cdots$ ,*pool_size* **do**                  `/* GradSign assisted part */`
        model = random_arch();
        model.score = GradSign(model);
        GradSign_pool.append(model);
    **end**
    arch = max_score(GradSign_pool) arch.acc = eval(arch) History.append(arch);
**end**
**return** max_acc(History);

---

---

**Algorithm 4:** G-REINFORCE

---

**Result:** Find the best performing architecture given the time constraint for 12000s

History = [];

pool_size;

policy $\pi_{\theta_0}$;

Reward = [];

baseline;

**while** *not exceeding time budget* **do**

    arch = generate_arch($\pi_{\theta_i}$);

    GradSign_pool = [];

    **for** $1, 2, \cdots$ ,*pool_size* **do**                `/* GradSign assisted part */`

        child = mutate_arch(arch);

        child.score = GradSign(child);

        GradSign_pool.append(child);

    **end**

    arch = max_score(GradSign_pool);

    arch.acc = eval(arch);

    r= arch.acc;

    History.append(arch);

    Reward.append(r);

    baseline.update(r);

    $\theta_{i+1} = \theta_i + \nabla_\theta \mathbb{E}_{\pi_{\theta_i}}[r - \text{baseline}]$

**end**

**return** max_acc(History);

**return**

---

---

**Algorithm 5:** G-HB

---

**Result:** Find the best performing architecture given the time constraint for 12000s

**Input:** budgets $b_{\min}$ and $b_{\max}$, $\eta$;

$s_{\max} = \lfloor \log_\eta \frac{b_{\max}}{b_{\min}} \rfloor$;

score_list = [];

pool_size;

**for** $s \in \{s_{\max}, s_{\max-1}, \cdots, 0\}$ **do**

    config_space = [];

    set $n = \lceil \frac{s_{\max}+1}{s+1} \cdot \eta^s \rceil$;

    **while** *sizeof(config_space) < n* **do**

        GradSign_pool = [];

        **for** $1, 2, \cdots$ ,*pool_size* **do**             `/* GradSign assisted part */`

            model = random_arch();

            **if** *model in score_list* **then**

                model.score = score_list[model];

            **else**

                model.score = GradSign(model);

                score_list[model] = model.score;

            **end**

            GradSign_pool.append(model);

        **end**

        arch = max_score(GradSign_pool);

        config_space.append(arch);

    **end**

    run SH on them with initial budget as $\eta^s \cdot b_{\max}$;

**end**

**return** best evaluated architecture;

---

