# OpenReview forum: "GradSign: Model Performance Inference with Theoretical Insights"
_ICLR.cc/2022/Conference — ICLR 2022 Poster_

### Official Review · Reviewer_YHPS · 2021-11-02

**Correctness:** 4
**Technical Novelty And Significance:** 3
**Empirical Novelty And Significance:** 3
**Recommendation:** 6
**Confidence:** 3

**Main Review:**

Paper is clearly written and experimental results are convincing. The idea of Gradsign is novel and integrating it to existing methods seems to yield improved performance.

I have some questions:

1) Does eq (2) have an intuitive meaning? Is so, can you pls explain it.
2) Table 1 shows that gradsign is task independent. Do the results generalise to other modalities such as text? Are there any benchmarks based on text datasets.

**Summary Of The Paper:**

This work discusses the problem of neural architecture search. While existing gradient methods are based on heuristics, this work proposes a metric called Gradsign for model performance inference, which provides some theoretical guarantees and performs well in practise.

Authors compare gradsign to a number of state of the art methods using 3 benchmarks involving CIFAR10, CIFAR100, and ImageNet16-120 datasets. Gradsign shows better performance compared to these methods.

**Summary Of The Review:**


The problem of automatically discovering efficient neural architectures is the foundation of work on AutoAI. From that perspective this is a nice and important piece of work that seems to have theoretical foundation and good practical performance.

---

> ### Author Response · Authors · 2021-11-19
> **Respond to reviewer YHPS**
>
> We thank the reviewer for the reviews and answer the main concerns below. We will also address all other comments in the final paper.
>
> `1. Does eq (2) have an intuitive meaning? If so, can you pls explain it?`
>
> Eq (2) is actually a mathematical formulation of Figure 1 when generalizing to multiple training samples and higher dimensions. More specifically, $\Psi$ is a quantity designated to measure the relative distance between local optima across training samples. Therefore, a smaller $\Psi$ indicates that the local optima of different samples are closer to each other, which further indicates a higher probability that different samples may have the same sign of gradients. In summary, Eq (2) formulates the negative correlation between the value of $\Psi$ and the probability of sampling gradients with the same sign across different samples. Section 3.1 in the revised paper also discusses the intuition behind Eq (2).
>
> `2. Table 1 shows that GradSign is task-independent. Do the results generalise to other modalities such as text? Are there any benchmarks based on text datasets?`
>
> Following the reviewer’s suggestion, we have found a NAS benchmark for NLP tasks ([6]) and compared GradSign with grad_norm and Synflow on the benchmark. The results are shown in Table 10 in the appendix. Although GradSign shows a stronger correlation to architectures’ accuracies compared to the baseline methods on this NLP task, the correlation score drops significantly compared to CV tasks. This could be potentially due to (1) text generation tasks or sequential models being more complex than CV tasks. Their setup may potentially violate some of our assumptions, eg. non-smooth loss landscape when doing sampling for text generation, and RNN cells have stronger second-order dependencies which make their Hessian highly non-diagonalizable; (2) the hyperparameters of the architecture candidates in the search space are not well-tuned in the benchmark dataset.
> We leave it as future work on designing effective gradient-based methods for NLP tasks.
>
> Table 10:
>
> |               | grad_norm | Synflow | GradSign |
> |---------------|-----------|---------|----------|
> | NAS-Bench-NLP | -0.11056  | 0.0546  | 0.1056   |
>
> reference:
>
> [6] Klyuchnikov, Nikita, et al. "NAS-Bench-NLP: neural architecture search benchmark for natural language processing." arXiv preprint arXiv:2006.07116 (2020)

---

### Official Review · Reviewer_CHXp · 2021-11-04

**Correctness:** 4
**Technical Novelty And Significance:** 3
**Empirical Novelty And Significance:** 3
**Recommendation:** 8
**Confidence:** 4

**Main Review:**

Overall, I am quite intrigued by the method. It's elegant and simple and seems to perform well. I do have some reservations around the execution and presentation of the work as listed below. That's why it's not a clear accept for me at this point.

### Strengths

* Well-written and natural to follow
* Interesting theoretical results and derivation. I don't enjoy reading through many theoretical results as they are often inaccessible. So I do appreciate the very clean and insightful presentation of the results.
* Good experimental performance (although missing a few comparisons potentially).

### Weaknesses

1. **NAS vs Pruning**: I think the authors mix weight pruning and NAS throughout the paper. This is really confusing and while pruning and NAS have lots of overlap I don't think it's appropriate for this paper. Their main contribution is a Model Performance Inference (MPI) score and as such much more suitable for the NAS framework (evaluating architectures for their performance on some target task).

2. **Pruning at initialization vs MPI scores**: (related to 1.) How can you compare against `SynFlow` and similar metrics that are designed for *weight pruning* at initialization? This seems like a strange comparison. I understand that the scores are compatible but could you provide some more context on why this is a useful comparison.

3. **Related work**: I think the authors need to include a few additional citations that they are missing. For example, most of recent advances in NAS are missing such as OFA ([Cai et al., 2019](https://arxiv.org/abs/1908.09791)), ProxylessNAS ([Cai et al., 2018](https://arxiv.org/abs/1812.00332)), MobileNetV3 ([Tan et al., 2019](http://proceedings.mlr.press/v97/tan19a.html)), EfficientNet ([Howard et al., 2019](http://openaccess.thecvf.com/content_ICCV_2019/html/Howard_Searching_for_MobileNetV3_ICCV_2019_paper.html)). [He et al., 2021](https://arxiv.org/pdf/1908.00709.pdf) also provide a useful survey for recent advances.

4. **Comparisons**: Two really important related papers that I believe merit a in-depth discussion:
   1. _[Deconstructing Lottery Tickets](https://arxiv.org/abs/1905.01067)_. They have some interesting results around the significance of the sign of the weights in the context of lottery tickets.
   2. _Zen-Nas ([Lin et al., 2021](http://openaccess.thecvf.com/content/ICCV2021/html/Lin_Zen-NAS_A_Zero-Shot_NAS_for_High-Performance_Image_Recognition_ICCV_2021_paper.html))_: Another really important related work to cite and compare against is Zen-NAS. It's an almost identical setup to your problem formulation.

5. **Experiments**: I think there are potentially two more interesting experiments for the authors to run.
    1. _[NATS-Bench](https://github.com/D-X-Y/NATS-Bench)_: The latest iteration of the NAS-Bench used by the authors.
    2. _ImageNet_: I think having one full-scale ImageNet experiment will be very powerful. The authors could even follow the setup of the Zen-NAS paper in order to potentially save on compute cost and recycle the results reported in that paper.

### Ways to improve my score

I do hope the authors decide to engage during the rebuttal period. I think the method has potential and hopefully, the authors can address the raised issues during that time.

---

### Update after the rebuttal

The authors engaged very productively with my feedback during the rebuttal and have addressed my concerns. The paper clearly meets the acceptance threshold now in my opinion. I am raising my overall score from 6 to 8 and "technical novelty" from 2 to 3.

**Summary Of The Paper:**

The paper proposes a new evaluation metric for scoring untrained, randomly initialized neural network architectures (zero-shot NAS) in order to predict their accuracy/performance after training. The score is based on evaluating the gradient signs and is shown to outperform existing approaches on NAS benchmarks (NAS-Bench 101, 201, NDS).

**Summary Of The Review:**

The paper proposes an interesting and sound idea but currently falls short of properly contextualizing their work within existing work and consequently comparing against recent work experimentally as well.

---

> ### Author Response · Authors · 2021-11-19
> **Respond to reviewer CHXp Part (1/2)**
>
> We thank the reviewer for the reviews and answer the main concerns below. We will also address all other comments in the final paper.
>
> `1. NAS vs Pruning`
>
> We agree that architecture pruning is a different task than NAS and should not be mixed with. Our current presentation and evaluation mainly include pruning techniques as additional heuristic baselines to compare GradSign against (we further explain our motivation in our answers to the next questions). We have updated the paper (modified places that may lead to confusion about NAS and pruning) to resolve the confusion.
>
> `2. Pruning at initialization vs MPI scores`
>
> In the revised paper, we have added a discussion on the relation between weight pruning at initialization and MPI to resolve the confusion. We agree that from a theoretical perspective weight pruning techniques (Snip, Grasp, Synflow) at initialization are not designated for MPI, however, they still have the potential to constitute decent baselines to compare GradSign with. By adding up all the weight-wise saliency scores (pruning at initialization scores) at initialization, the method could easily be adapted to heuristic methods for MPI which has been empirically verified and included in several MPI works [1][4][5]. Besides pruning based heuristic methods, our baselines also include recent works specifically designed for MPI (NASWOT and ZenNAS).
>
> `3. Related work`
>
> We thank the reviewer for bringing up additional related works and have included them in the revised paper.
>
> `4. 1 Comparison: Deconstructing Lottery Tickets`
>
> The sign of weights mentioned in this paper (empirically observed) has no theoretical connections to the sign of gradients besides a hypothesis that the sign of weights can largely preserve the sign of gradients in back-propagation, which we cannot prove theoretically. However, this is an interesting and promising research direction to dive further, and we leave this as future work. We thank the reviewer for bringing this up.
>
> `4. 2 Comparison: ZenNAS`
>
> Our revised paper includes a comprehensive comparison against ZenNAS. First, section A.3 in the appendix presents the design difference between GradSign and ZenNAS. Second, we include an additional experiment (Table 3) that compares Zen scores with GradSign on NAS-Bench-201 using CIFAR-10, CIFAR-100, and ImageNet16-120. In addition, we also include ImageNet-1k results (Figure 4) comparing GradSign with ZenNAS in the appendix.
>
> Table 3:
>
> | Dataset        | ZenNAS | grad_norm |  snip | grasp | fisher | Synflow | NASWOT | GradSign |
> |----------------|:------:|:---------:|:-----:|:-----:|:------:|:-------:|:------:|:--------:|
> | CIFAR10        | -0.016 |   0.594   | 0.595 |  0.51 |  0.36  |  0.737  |  0.728 |   0.765  |
> | CIFAR100       | -0.041 |   0.637   | 0.637 | 0.549 |  0.386 |  0.763  |  0.703 |   0.793  |
> | ImageNet16-120 |  0.032 |   0.579   | 0.579 | 0.552 |  0.328 |  0.751  |  0.696 |   0.783  |
>
> `5.1 Experiments: NATS-Bench`
>
> We thank the reviewers for the suggestion. Table 8 and Table 9 in the revised paper shows our experimental results on NATS-Bench. We include both correlation evaluation of GradSign through Spearman’s rho and searching results from GradSign assisted NAS algorithms on NATS-Bench, which verify that GradSign can achieve similar performance on the latest NATS-Bench as well.
>
> Table 8:
>
> |               | CIFAR-10 | CIFAR-100 | ImageNet16-120 |
> |---------------|:--------:|:---------:|:--------------:|
> | NAS-Bench-201 |   0.765  |   0.793   |      0.783     |
> | NATS-Bench    |   0.760  |   0.792   |      0.784     |
>
> Table 9:
>
> |   Methods   |   CIFAR-10  |             |  CIFAR-100  |             | ImageNet16-120 |             |
> |:-----------:|:-----------:|:-----------:|:-----------:|:-----------:|:--------------:|:-----------:|
> |             |  Validation |     Test    |  Validation |     Test    |   Validation   |     Test    |
> |     REA     | 91.06+-0.49 | 93.84+-0.45 | 71.53+-1.31 | 71.60+-1.27 |   44.82+-1.23  | 45.18+-1.37 |
> |    G-REA    | 91.35+-0.35 | 94.15+-0.32 | 72.67+-1.05 | 72.65+-0.97 |   45.55+-0.96  | 45.99+-0.93 |
> |      RS     | 90.95+-0.28 | 93.77+-0.26 | 71.01+-0.97 | 71.15+-0.95 |   44.58+-0.95  | 44.73+-1.10 |
> |     G-RS    | 91.23+-0.22 | 94.02+-0.22 | 72.12+-0.82 | 72.15+-0.78 |   45.43+-0.74  | 45.83+-0.80 |
> |  REINFORCE  | 90.92+-0.38 | 93.71+-0.37 | 71.04+-1.02 | 71.17+-1.12 |   44.56+-0.97  | 44.80+-1.18 |
> | G-REINFORCE | 91.20+-0.23 | 93.98+-0.23 | 71.93+-0.91 | 72.05+-0.89 |   45.28+-0.77  | 45.64+-0.86 |

---

> > ### Author Response · Authors · 2021-11-19
> > **Respond to reviewer CHXp Part (2/2)**
> >
> > `5.2 Experiments: ImageNet`
> >
> > Zen-NAS requires 480 epochs to train selected architectures and 480,000 evolution iterations for choosing the architecture, which runs for nearly one month on a V100 GPU. Due to the time constraint, we compare the architectures selected using the Zen and GradSign scores on the Imagenet dataset by letting both methods run 1,000 evolution iterations to select an architecture and train the architecture for 20 epochs. The results are shown in Figure 4 in the appendix. We will evaluate Zen and GradSign using the experimental setup in [4] (480K evolution iterations and 480 training epochs) in the final paper.
> >
> > references:
> >
> > [1] Abdelfattah, Mohamed S., et al. "Zero-cost proxies for lightweight nas." arXiv preprint arXiv:2101.08134 (2021).
> >
> > [4] Lin, Ming, et al. "Zen-NAS: A Zero-Shot NAS for High-Performance Image Recognition." Proceedings of the IEEE/CVF International Conference on Computer Vision. 2021.
> >
> > [5] Mellor, Joe, et al. "Neural architecture search without training." International Conference on Machine Learning. PMLR, 2021.

---

> ### Comment · Reviewer_CHXp · 2021-11-29
> **Great rebuttal**
>
> Thank you for the detailed responses.
>
> I very much appreciate the updated experiments and results. I think the simplicity of the approach combined with the very decent experimental results definitely makes for a good contribution to ICLR.
>
> The comparison to ZenNAS is very interesting and I agree with the authors that their approach will be much easier to generalize to other tasks/architectures/datasets compared with ZenNAS.
>
> Hopefully, the authors will follow up with their promise to include a more thorough comparison in the final revision. I think this will strengthen the contribution and increase the potential impact of the paper.

---

> > ### Author Response · Authors · 2021-11-30
> > **Thanks for the updates**
> >
> > We really thank the reviewer for the effort and time in reviewing as well as giving insightful suggestions.
> >
> > We will surely include the complete comparison in the final paper.

---

### Official Review · Reviewer_H9Ev · 2021-11-08

**Correctness:** 3
**Technical Novelty And Significance:** 3
**Empirical Novelty And Significance:** 2
**Recommendation:** 6
**Confidence:** 2

**Main Review:**

Strengths:
1. The proposed metric is efficient to compute. It only requires the gradient information of a minibatch at a random initialization point.
2. The underlying theory is sound. The proposed metric is backed by theoretical insights.

Weaknesses:
1. In the evaluation, the paper only compares GradSign with gradient-based approaches. I would like to see a comparison against other sample-based, learning-based approaches as well. You can compare the prediction accuracy and time cost of each method and show the trade-off. For example, a sample-based approach can be more accurate but takes 10x more time than GradSign.

Additional questions:
1. How does the hyperparameter of training (e.g., learning rate) affect the prediction accuracy of GradSign?

**Summary Of The Paper:**

Model performance inference is a key challenge in neural architecture search. This paper introduces GradSign, an accurate, simple, and flexible metric for model performance inference. GradSign approximately analyzes the optimization landscape of different networks at the granularity of individual training samples using the gradients evaluated at a random initialization state.
Experimental results show that GradSign can generalize well to real-world networks and outperform state-of-the-art gradient-based methods for model performance inference.

**Summary Of The Review:**

This paper introduces a simple yet effective way for model performance inference. It outperforms existing gradient-based approaches in almost all experiments. I recommend accepting this paper.

---

> ### Author Response · Authors · 2021-11-19
> **Respond to reviewer H9Ev**
>
> We thank the reviewer for the reviews and answer the main concerns below. We will also address all other comments in the final paper.
>
> `1. In the evaluation, the paper only compares GradSign with gradient-based approaches. I would like to see a comparison against other sample-based, learning-based approaches as well. You can compare the prediction accuracy and time cost of each method and show the trade-off. For example, a sample-based approach can be more accurate but takes 10x more time than GradSign.`
>
> Given the time limit during the discussion period, we mainly collected experimental results for learning-based and sample-based methods from related works with similar experimental setups. These comparisons are now available in Appendix A.3 of the revised manuscript.
>
> Sample-based: According to [1], one of the state-of-the-art sample-based methods EcoNAS achieves a similar Spearman correlation score compared to GradSign on CIFAR-10 by using ~ 600 mini-batches whereas GradSign only needs one mini-batch. EcoNAS+ is able to achieve higher correlation scores by using more mini-batches, which eventually converges to 0.85.
>
> Learning-based: [2] has conducted several experiments on evaluating learning-based methods using MLP, LSTM and GNN. We compare GradSign with these learning-based approaches on the NAS-Bench-201 dataset. To achieve a similar Kendall score as GradSign, MLP, LSTM, and GATEs require approximately 800, 400, and 100 well-trained candidate architectures as their training datasets, respectively. It generally requires 200 epochs (~40K mini-batches according to the setup in [3]) to train a candidate architecture from scratch. Due to the time limit, we were not able to fully reproduce their results from the codebase, however, we plan to include a more thorough comparison against learning-based methods in the final paper. We include the current results in Table 7.
>
> Table 7:
>
> |          | Kendall's Tau | Average mini-batches per sample |
> |:--------:|:-------------:|:------------------------------:|
> |    MLP   |     0.5388    |              1959              |
> |   LSTM   |     0.6407    |               978              |
> |  GATES-1 |      0.45     |              1959              |
> |  GATES-2 |     0.7401    |               195              |
> | GradSign |     0.6016    |                1               |
>
> `2. How does the hyperparameter of training (e.g., learning rate) affect the prediction accuracy of GradSign?`
>
> Our theoretical analysis is based on an assumption that each candidate architecture is trained to be near-optimal in order to satisfy our theoretical derivation; an inappropriate learning rate (or other hyperparameters) can largely harm the predictive performance of the architecture. In the worst case, an unreasonable learning rate can result in a significant accuracy drop for even promising architectures, which cannot be covered by the GradSign score. However, as long as the learning rate is fairly appropriate to guarantee convergence to near-optimal weights, the GradSign’s performance can be largely preserved. To verify this, Table 8 and Table 9 in the revised paper compare the GradSign scores on benchmarks with different hyperparameters (i.e., NAS-Bench-201 and NATS-Bench). The results show that GradSign is robust to different selections of hyperparameters.
>
> Table 8:
>
> |               | CIFAR-10 | CIFAR-100 | ImageNet16-120 |
> |---------------|:--------:|:---------:|:--------------:|
> | NAS-Bench-201 |   0.765  |   0.793   |      0.783     |
> | NATS-Bench    |   0.760  |   0.792   |      0.784     |
>
> Table 9:
>
> |   Methods   |   CIFAR-10  |             |  CIFAR-100  |             | ImageNet16-120 |             |
> |:-----------:|:-----------:|:-----------:|:-----------:|:-----------:|:--------------:|:-----------:|
> |             |  Validation |     Test    |  Validation |     Test    |   Validation   |     Test    |
> |     REA     | 91.06+-0.49 | 93.84+-0.45 | 71.53+-1.31 | 71.60+-1.27 |   44.82+-1.23  | 45.18+-1.37 |
> |    G-REA    | 91.35+-0.35 | 94.15+-0.32 | 72.67+-1.05 | 72.65+-0.97 |   45.55+-0.96  | 45.99+-0.93 |
> |      RS     | 90.95+-0.28 | 93.77+-0.26 | 71.01+-0.97 | 71.15+-0.95 |   44.58+-0.95  | 44.73+-1.10 |
> |     G-RS    | 91.23+-0.22 | 94.02+-0.22 | 72.12+-0.82 | 72.15+-0.78 |   45.43+-0.74  | 45.83+-0.80 |
> |  REINFORCE  | 90.92+-0.38 | 93.71+-0.37 | 71.04+-1.02 | 71.17+-1.12 |   44.56+-0.97  | 44.80+-1.18 |
> | G-REINFORCE | 91.20+-0.23 | 93.98+-0.23 | 71.93+-0.91 | 72.05+-0.89 |   45.28+-0.77  | 45.64+-0.86 |

---

> > ### Author Response · Authors · 2021-11-19
> > **Respond to reviewer H9Ev, References**
> >
> > References:
> >
> > [1] Abdelfattah, Mohamed S., et al. "Zero-cost proxies for lightweight nas." arXiv preprint arXiv:2101.08134 (2021).
> >
> > [2] Ning, Xuefei, et al. "A generic graph-based neural architecture encoding scheme for predictor-based nas." Computer Vision–ECCV 2020: 16th European Conference, Glasgow, UK, August 23–28, 2020, Proceedings, Part XIII 16. Springer International Publishing, 2020.
> >
> > [3] Dong, Xuanyi, and Yi Yang. "Nas-bench-201: Extending the scope of reproducible neural architecture search." arXiv preprint arXiv:2001.00326 (2020).

---

### Author Response · Authors · 2021-11-19
**General respond**

We thank the reviewers for all their constructive comments. Based on the comments (please refer to our individual replies for details), we have prepared a revised submission, where we focus on clarifying the most important confusions we identified and providing additional results as much as possible. All revised texts are in blue. Most of the additional experimental results are included in Appendix A.3.

Summary:
1. Added learning-based and sample-based results for trade-off comparison.
2. Removed several sentences which might bring confusion about the relationship between NAS and pruning in the introduction section. Added clarification of the connection between pruning at initialization and model performance inference (MPI).
3. Added several related works mentioned by the reviewers.
4. Added ZenNAS as a baseline for Spearman's $\rho$ evaluation on NAS-Bench-201 as well as a comparison of GradSign and ZenNAS on ImageNet-1k.
5. Additional results for GradSign on the latest version of NAS-Bench-201 (NATS-Bench).
6. Additional results on NAS-Bench-NLP.

---

### Decision · Program_Chairs · 2022-01-20

**Decision:**

Accept (Poster)

**Comment:**

The paper is focussed on proposing a new evaluation metric for evaluating untrained, randomly initialized neural network architectures towards predicting their accuracy/performance after training. The metric they propose is based on evaluating the gradient sign. The method shown to outperform existing approaches on NAS benchmarks.

The reviewers found the paper's idea simple but effective. The experimental evaluation and efficacy of the proposed method were the main strong points of the paper. The paper was also significantly improved during the discussion period both in terms of presentation and the scope of experiments/comparisons was enlarged.

While the metric is theoretically motivated, I personally found some of the theoretical statements weak in terms of assumptions/clarity. I would request the authors to consider taking this and other suggestions made by reviewers into account

Overall I recommend acceptance based on the strong and thorough experimental results shown by the paper on a problem of clear interest to the community.